# Interpreting Differentiable Latent States for Healthcare Time-series Data

**Yu Chen** [1]  **Nivedita Bijlani** [2]  **Samaneh Kouchaki** [2]  **Payam Barnaghi** [1]

## Abstract

Machine learning enables extracting clinical insights from large temporal datasets. The applications of such machine learning models include identifying disease patterns and predicting patient outcomes. However, limited interpretability poses challenges for deploying advanced machine learning in digital healthcare. Understanding the meaning of latent states is crucial for interpreting machine learning models, assuming they capture underlying patterns. In this paper, we present a concise algorithm that allows for i) interpreting latent states using highly related input features; ii) interpreting predictions using subsets of input features via latent states; and iii) interpreting changes in latent states over time. The proposed algorithm is feasible for any model that is differentiable. We demonstrate that this approach enables the identification of a daytime behavioral pattern for predicting nocturnal behavior in a real-world healthcare dataset.

## 1. Introduction

Digital healthcare is a rapidly growing field empowered by the internet, computers, and mobile devices. It offers promising prospects for enhancing healthcare quality and accessibility. Machine learning techniques have the potential to unlock valuable insights from extensive temporal datasets, enabling healthcare providers to identify disease patterns and predict patient outcomes. However, deploying sophisticated machine learning technologies in digital healthcare poses challenges due to their limited interpretability. The interpretability of machine learning models is crucial in healthcare applications because it enables professionals to understand how a model arrived at a particular prediction or decision. This is especially important when making decisions that can have significant consequences for patients, such as in disease diagnosis or treatment recommendation.

Explainable Artificial Intelligence (XAI) has drawn increasing attention in recent years due to the growing use of deep learning models in critical decision-making applications. Various approaches have been developed for explaining black-box models, such as feature importance explanations (Datta et al., 2016; Ribeiro et al., 2016; Lundberg & Lee, 2017; Sundararajan et al., 2017) and example-based explanations (Li et al., 2018; Chen et al., 2019; Gurumoorthy et al., 2019; Crabbé et al., 2021). However, when it comes to healthcare applications, obtaining only the importance of each feature or example is not enough for interpreting the underlying patterns. Understanding the meaning of latent states (representations) learned by a black-box model is key to this issue, assuming they capture underlying patterns. We posit that a latent state can be translated to a comprehensible pattern that is formed by a subset of features.

In this paper, we introduce a simple, yet efficient algorithm that aims to: i) interpret latent states using highly related input features; ii) interpret predictions using subsets of input features via latent states; and iii) interpret changes in latent states over time. Once the first two goals are achieved, the algorithm can readily fulfill the third one as long as the model is able to learn latent states over time. This algorithm is feasible for any differentiable latent state with respect to input features. Hence, it can work as a readily available plugin for most deep-learning models for temporal data, such as transformers (Vaswani et al., 2017), recurrent neural networks (Tealab, 2018), Long Short-Term Memory (LSTM) networks (Hochreiter & Schmidhuber, 1997), and advanced state space models (Gu et al., 2021). To demonstrate the portability of our method, we additionally provide a neat solution for applying it to models using Neural Controlled Differential Equation (NCDE) (Kidger et al., 2020). NCDE models are capable of i) fitting the dynamics of latent states (i.e. first-order derivatives w.r.t. time), ii) capturing long-term dependencies in latent states, iii) dealing with irregularly sampled time series, and iv) working in an online fashion. In short, NCDE is an attractive approach for modeling healthcare time-series data. We demonstrate that the proposed algorithm enables the identification of a daytime behavioral pattern for predicting nocturnal behavior in a real-world healthcare dataset using a NCDE model.

[1]Department of Brain Sciences, Imperial College London, UK [2] Department of Electrical and Electronic Engineering, University of Surrey, UK. Correspondence to: Yu Chen <yu.chen@imperial.ac.uk>.

*Workshop on Interpretable ML in Healthcare at International Conference on Machine Learning (ICML)*, Honolulu, Hawaii, USA. 2023. Copyright 2023 by the author(s).

## 2. Related work

In this section, we introduce several prior methods in XAI that are related to our work.

### 2.1. Feature importance to predictions

The most popular methods in XAI focus on identifying important features for the prediction of a given sample, such as SHAP (Lundberg & Lee, 2017) and LIME (Ribeiro et al., 2016). These methods do not consider the factor of time. Crabbé et al. (2021) proposed Dynamask to identify feature importance over time, and Crabbé & van der Schaar (2022) proposed for measuring feature importance without labels. However, we aim to detect salient patterns rather than individual features, which could be more helpful in understanding disease progression or predicting certain symptoms.

### 2.2. Interpreting latent states

We can view a latent state as a compact code of an underlying pattern of observed features. It is often referred to as a latent representation. Esser et al. (2020) introduced an approach to transform a latent state into an interpretable one. The interpretability comes from a decomposition of semantic concepts, which requires prior knowledge to find associated semantic concepts in the data. Accessing this information is rather difficult in healthcare data.

The integrated Jacobian (also known as integrated gradients) (Sundararajan et al., 2017; Crabbé et al., 2021) can quantify the impact of an input feature on the shift of a latent state. Although this is a one-to-one impact measure, we can utilize it to identify subsets of key features for each latent state.

Assume we have a model $\mathcal{G}$ to infer the latent state $z \in R^H$ of an observation $x \in R^D$, i.e. $z = \mathcal{G}(x) : R^D \to R^H, \; H < D$. Let $x_i$ denote the $i$-th feature of a baseline sample $x$, $\hat{x}_i$ denotes the $i$-th feature of a test sample $\hat{x}$. We can compute the integrated Jacobian of the $i$-th feature $\mathbf{j}_i \in R^H$ regarding $(x, \hat{x})$ as below:

$$\mathbf{j}_i = (x_i - \hat{x}_i) \int_0^1 \frac{\partial \mathcal{G}(x)}{\partial x_i}\Big|_{\gamma_i(\lambda)} d\lambda, \tag{1}$$

Here $\gamma_i(\lambda) = \hat{x}_i + \lambda(x_i - \hat{x}_i), \; \forall \lambda \in [0, 1]$, represents a point between $x_i$ and $\hat{x}_i$.

## 3. Methods

In this section, we first define a measurement that quantifies the impact of an input feature on a latent state. We then propose an algorithm for identifying the most influential features that affect each latent state. With this algorithm, we can interpret each latent state by two sets of influential features, which likely result in the most positive and negative magnitude shift of a latent state. We can further analyze

how these feature sets affect predictions by combining them with other methods (such as SHAP, LIME).

### 3.1. Impact of an input feature to a latent state

We quantify the impact of $i$-th input feature on $s$-th latent state in terms of the shift between two samples $(x, \hat{x})$ as the following equation:

$$p_{i,s} = \frac{\mathbf{j}_{i,s}}{|z_s - \hat{z}_s|}, \; z = \mathcal{G}(x), \; \hat{z} = \mathcal{G}(\hat{x}) \tag{2}$$

$\mathbf{j}_{i,s}$ is the $s$-th element in $\mathbf{j}_i$ which can be obtained by Equation (1). $z_s, \hat{z}_s$ are the $s$-th element in $z$ and $\hat{z}$ respectively.

The absolute value of $p_{i,s}$ is the ratio of the shift caused by the specific feature compared to the total shift caused by all features for a latent state. The impact can be in the positive (increasing) or negative (decreasing) direction which enables us to analyze the shifts more precisely. Crabbé et al. (2021) provides a measurement called projected Jacobian which quantifies the total impact of an input feature on all latent states. Using Equation (2), we can measure the impact of an input feature on individual latent states.

### 3.2. Interpreting latent states by influential features

---

**Algorithm 1** Generate Feature Heat Map of Latent States

---

**Input:** The direction of impact $d \in \{True, False\}$, training set $\{X, y\}$, model $\mathcal{G}$, baseline set $\hat{X}$, the direction of impact $d \in \{True, False\}$, number of latent states $H$, number of input features $D$, size of a subset of training samples $m$, number of top samples $k$, number of latent states $h$, and number of top features $l$.

$X_c \leftarrow$ balanced subset$(X, y, m)$
$Z_c \leftarrow$ latent states$(X_c, \mathcal{G})$
Initialize the heat map matrix $\mathbf{M}$ with zeros: $\forall \; 0 \leq i < H, 0 \leq j < D, \mathbf{M}_{i,j} = 0$
**for** $\hat{x} \in \hat{X}$ **do**
  $\hat{z} \leftarrow$ latent states$(\hat{x}, \mathcal{G})$
  $\mathbf{Z}_{id} \leftarrow$ TopDissimilarSamples$(Z_c, \hat{z}, k)$
  $\mathbf{F} \leftarrow$ TopImpactfulFeatures$(d, \mathcal{G}, \hat{x}, X_c[\mathbf{Z}_{id}], l)$
  **for** $k_i \in \{1, \dots, k\}$ **do**
    **for** $h_i \in \{1, \dots, h\}$ **do**
      $i = h_i, \; \forall j \in \mathbf{F}_{k_i,h_i}, \; \mathbf{M}[i, j] + = 1$
    **end for**
  **end for**
**end for**
**Return:** $\mathbf{M}$

---

Our proposed approach for interpreting latent states is based on contrastive methods, i.e. at least one pair of different samples is required to compute the similarities and shifts in latent and input spaces.

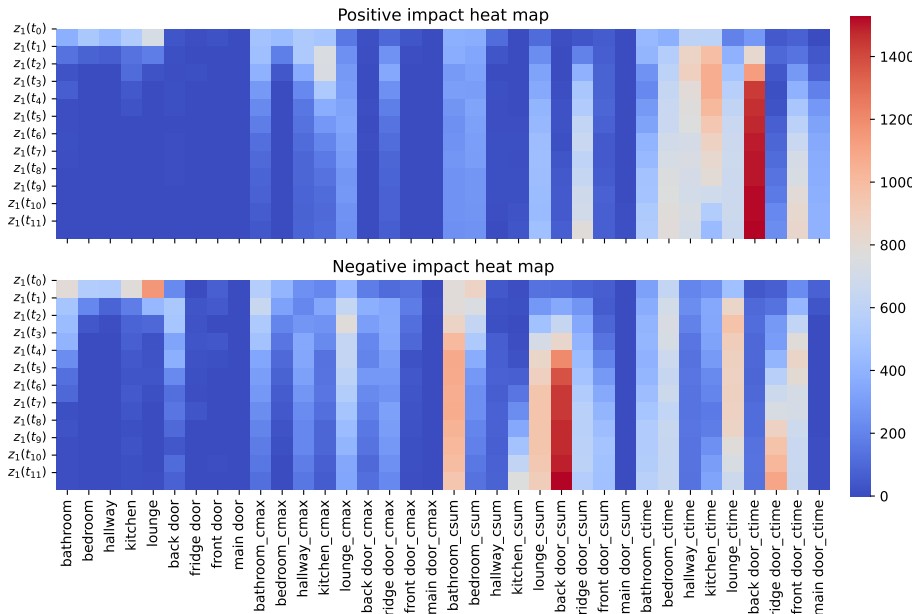

*Figure 1.* The two heat maps for interpreting latent state $z_1$ over time $t$ learned by the NCDE model. The y-axis is the latent state along the time axis, $z_1(t_n)$ means $z_1$ at the $n$-th time step. The x-axis is the input features which are named by the locations of sensors and the aggregate methods, e.g. "kitchen" is the number of activities in the kitchen during one hour, "kitchen_ctime" is the cumulative number of hours that "kitchen" has non-zero values, "kitchen_cmax" is the maximum value of "kitchen" among all passed hours, etc.

More specifically, we identify the most different samples $\{x|x \in X_{dif}\}$ compared to a baseline sample $\hat{x}$ and locate features that contribute most to the positive/negative shift of each latent state between each pair of $(x, \hat{x})$. To obtain an overall result, we iterate this process on a small set of baseline samples and generate a heat map that indicates how frequently an input feature has been identified as one of the most impactful features of a latent state. Please refer to Algorithms 1 to 3 for more details. By visualizing this heat map, we can readily observe the relation between latent states and input features over time. Figure 1 shows an example of this heat map obtained by real-world data.

In order to identify influential features, we focus on the most dissimilar pairs of training samples. These pairs are likely to exhibit significant shifts in their latent states. On the other hand, the most similar pairs tend to have minimal shifts in most latent states, making it challenging to determine which features contribute the most to their similarity.

---

**Algorithm 2** TopDissimilarSamples

**Input:** Latent states of the subset samples $Z$, latent states of a baseline sample $z$, number of top samples $k$.

$S_{cos} \leftarrow$ Cosine Similarities between $z$ and $Z$
$S_{ids} \leftarrow$ Argsort($S_{cos}$, ascending=True)
**Return:** $S_{ids}[: k]$

---

**Algorithm 3** TopImpactfulFeatures

**Input:** The direction of impact $d \in \{True, False\}$, a differentiable model $\mathcal{G}$, input features of a baseline sample $\hat{x} \in R^D$, input features of a subset samples $X \in R^{k \times D}$, number of top features $l$.

$P \leftarrow$ ImpactMeasure($d,X,\hat{x},\mathcal{G}$)    # apply Equation (2) on all pairs of $\{(\hat{x}, x)|x \in X\}$ for each latent state.
$P_{id} \leftarrow$ Argsort($P$, descending=$d$, axis=-1)
**Return:** $P_{id}[:,:,:l]$

---

### 3.3. Interpreting latent states of NCDE models

Assuming that there are latent states representing the underlying patterns of time series, which depend on the cumulative influence from all previous states, differential equations are often applied to find the dynamics of such latent states and forecast future states.

3.3.1. PRELIMINARY: NEURAL CONTROLLED DIFFERENTIAL EQUATION (NCDE)

Neural Ordinary Differential Equation (NODE) (Chen et al., 2018) is a deep learning technique that can use neural networks to approximate a latent state that has no explicit formulation with its dynamics. For instance: $z(t_1) = z(t_0) + \int_{t_0}^{t_1} f_\theta(z(t),t)dt$. Here $z(t)$ denotes latent states at time $t$, $f_\theta(\cdot)$ represents an arbitrary neural network. NCDE

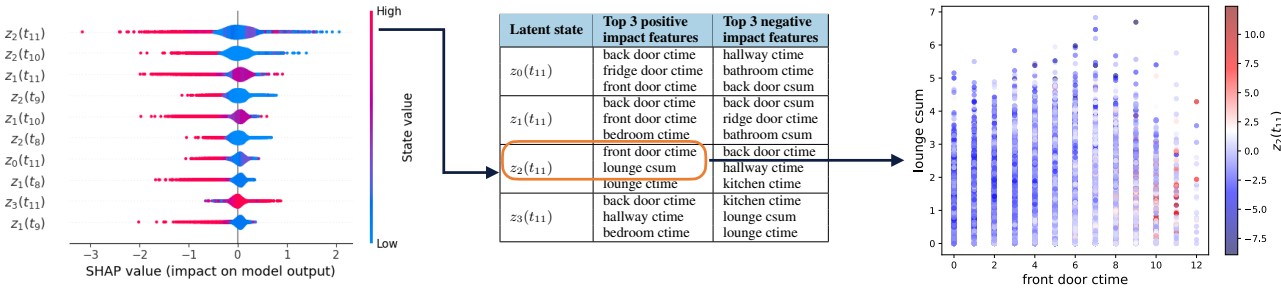

*Figure 2.* A demonstration of identifying a behavioral pattern that is likely related to the model output. We first identify the most important latent state $z_2(t_{11})$ of the model output by SHAP value (the top one in the left figure); we then get the top influential features of the top latent state (the third row in the table) by the proposed method; finally, we can analyze the correlations between the specific latent state and its top features, e.g. using scatter plot to show how the top two features with positive impact correlate to the value of the latent state (the right figure). The example shows that it is more likely to get higher values of $z_2(t_{11})$ when *lounge cum* (the cumulative sum of activities in the lounge) is low and *front door ctime* (the cumulative active hours at the front door) is high. Higher values of $z_2(t_{11})$ are likely to give a negative impact on the model output, which indicates less likely being awake during night time.

(Kidger et al., 2020) uses a controlled different equation, in which the dynamics of $z(t)$ are assumed to be controlled or driven by an input signal $x(t)$.

$$z(t_1) = z(t_0) + \int_{t_0}^{t_1} \tilde{f}_\theta(z(t), t) dx(t)$$
$$= z(t_0) + \int_{t_0}^{t_1} \tilde{f}_\theta(z(t), t) \frac{dx}{dt} dt, \quad (3)$$

where $\tilde{f}_\theta : R^H \rightarrow R^{H \times D}$, $H$ is the dimension of $z$, $D$ is the dimension of $x$. The NODE is a special case of NCDE when $dx/dt = I$. By treating $f_\theta(z(t), t)\frac{dx}{dt}$ as one function $g_\theta(z(t), t, x)$, one can solve NCDE by regular methods for solving NODE. The trajectory of control signals $(x(t))$ in NCDE can be fitted independently with $\tilde{f}_\theta$, which enables NCDE to deal with irregular sampled time series and work with online streaming data.

### 3.3.2. INTEGRATED JACOBIAN OF NCDE MODELS

We cannot compute the integrated Jacobian of NCDE models readily by Equation (1) because $z \neq \mathcal{G}(x)$ in this case. Interestingly, we can obtain the Jacobian $\partial z/\partial x$ easily when we i) use observations as control signals; ii) $x(t)$ is approximated by an invertible function. According to the chain rule:

$$\frac{\partial z}{\partial x} = \frac{\partial z}{\partial t}\frac{dt}{dx} = \left(\tilde{f}_\theta(z(t), t)\frac{dx}{dt}\right)\frac{dt}{dx} = \tilde{f}_\theta(z(t), t) \quad (4)$$

This way only needs a forward computation instead of a back-propagation to obtain the Jacobian $\partial z/\partial x$.

## 4. Experiments

We conducted experiments on two different datasets: 1). one uses data from an ongoing study of dementia care; 2)

the other is the Sepsis dataset (Reyna et al., 2019) from PhysioNet [1]. We use NCDE models for both datasets. We also conducted feature augmentation by cumulative operations on both datasets. For example: "feature_ctime" is the cumulative number of time steps that a specific feature has non-zero values, "feature_cmax" is the maximum value of a specific feature among all passed time steps, "feature_csum" is the value sum of a specific feature over passed time steps. Each time step represents one-hour information for both datasets.

### 4.1. Experiments with dementia care data

The data were collected from sensors deployed in 91 participants' homes, including Passive Infra-Red (PIR) sensors installed in multiple locations, door sensors, and an under-the-mattress sleep mat. The dataset is not public but a smaller dataset with similar properties is available on Zenodo repository [2]. We train an NCDE model using household sensor data between 6:00 AM and 6:00 PM daily to predict if a participant will be awake more than half of the time when he/she is in bed during the night. Each data sample includes 12 time steps each of which has 36 features that are generated from sensor activations and feature augmentation.

The NCDE model learns four latent states at each time step and we concatenate all latent states as the input to the final linear layer. Figure 1 demonstrates the heat maps for interpreting one of these latent states at each time step. It illustrates how this latent state evolves over time. Combined with SHAP values (Lundberg & Lee, 2017) of top 10 important latent states (the left figure in Figure 2), we can discover

[1]https://physionet.org/content/challenge-2019/1.0.0/
[2]https://zenodo.org/record/7622128

subsets of features that jointly affect the model prediction through latent states. In Figure 2, we demonstrate how to identify a behavioral pattern using the aforementioned dataset by the proposed method. The figure shows that it is more likely to get higher values of $z_2(t_{11})$ when *lounge cum* (the cumulative sum of activities in the lounge) is low and *front door ctime* (the cumulative active hours at the front door) is high. Higher values of $z_2(t_{11})$ are likely to give a negative impact on the model output, which indicates less likely being awake during night time.

### 4.2. Experiments with Sepsis dataset

The Sepsis dataset (Reyna et al., 2019), applied in the Physi-oNet/Computing in Cardiology Challenge 2019, is a public dataset consisting of hourly vital signs and lab data, along with demographic data, for 40366 patients obtained from three distinct U.S. hospital systems. The objective is to predict sepsis within 72 hours from the time a patient was admitted to the ICU. We use a subset of this dataset which consists of data from 12000 patients. The task is a binary classification of predicting the onset of sepsis. Therefore, each data sample includes 72 time steps (hours) each of which has 136 features that are augmented by 34 original features (vital signs and lab data). For patients who have no records of the full 72 hours, we treated features of missing hours as missing values filled by 0. All original features were preprocessed by min-max normalization and shifted from the range [0,1] to the range [1,2].

The NCDE model learns two latent states at each time step. The SHAP values plot in Figure 3(a) shows the 10 most impactful latent states on the sepsis outcome. $z_0(t_{35})$ is the most influential latent state according to the figure, which is the first latent state at the 36-th hour. Using our method to link this latent state to the most influential input features, the top three on $z_0(t_{35})$ are *SBP_csum*, *O2sat_csum*, and *Resp_csum*. We see from Figures 3(b) and 3(c) that when values of *SBP_csum* (cumulative sum of Systolic BP), *O2sat_csum* (cumulative sum of Pulse oximetry), and *Resp_csum* (cumulative sum of Respiration rate) are all in the lower side, $z_0(t_{35})$ is likely having a larger value and thus likely having a positive impact on the prediction, i.e., increasing the likelihood of sepsis.

## 5. Conclusion and future work

We propose a method that can identify the most influential features of a latent state, which may or may not be linearly correlated. Our method allows linking model outcomes to the most contributing features, which brings insights into understanding the knowledge that learned by the model. However, after discovering influential features, we still need to manually analyze the relations between those features, latent states, and model outputs. In order to get more readily

interpretable results, we consider developing algorithms that can automatically discover interpretable patterns using top influential features in the future. More future work can be done in multiple directions, such as causal analysis of identified features, or probabilistic modeling on joint distributions of those features.

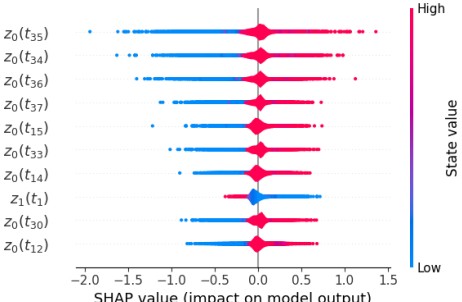

(a) SHAP values of top 10 latent states on Sepsis dataset

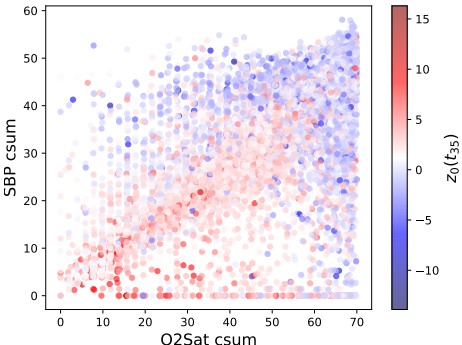

(b) Scatter plot using values of *SBP_csum*, *O2sat_csum* at $t_{35}$, the color indicates values of the top latent state $z_0(t_{35})$

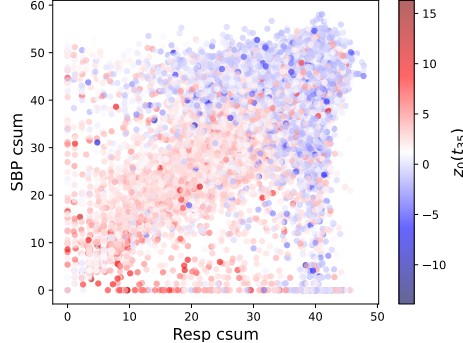

(c) Scatter plot using values of *SBP_csum*, *Resp_csum* at $t_{35}$, the color indicates values of the top latent state $z_0(t_{35})$

*Figure 3.* Interpreting the top latent state in Sepsis dataset

# Acknowledgment

This project was supported by the Engineering and Physical Sciences Research Council (EPSRC) PROTECT Project (grant number: EP/W031892/1) and the UK DRI Care Research and Technology Centre funded by the Medical Research Council (MRC) and Alzheimer's Society (grant number: UKDRI-7002).

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
