# OpenReview forum: "Interpreting Differentiable Latent States for Healthcare Time-series Data"
_ICML.cc/2023/Workshop/IMLH — IMLH 2023 Poster_

### Official Review · Reviewer_auNS · 2023-06-13
**This paper proposes an algorithm for interpreting latent states, interpreting predictions, and interpreting changes in latent states over time using household sensor data.**

**Rating:** 7
**Confidence:** 4

**Review:**

The NEURAL CONTROLLED DIFFERENTIAL EQUATION (NCDE) model was chosen to represent the dynamics of the latent states. The model was designed to predict if a participant will be awake more than half of the time when he/she is in bed during the night.

1. This work is significant in the area of interpretability for both supervised and unsupervised models.
2. This paper quantifies the impact of a feature on a latent state.
3.  It identifies the set of input features for each latent state using integrated gradients.
4. This work is very similar to the following in their goals, will be good referencing it:\
*Crabbé, Jonathan, and Mihaela van der Schaar. "Label-Free Explainability for Unsupervised Models." Proceedings of Machine Learning Research (2022).*
5. SHAP values have limitations in determining feature importance. Relying completely on SHAP may mislead choosing the top important latent states.

---

### Official Review · Reviewer_vbBX · 2023-06-15
**This paper proposes a method to identify the most influential features of a latent state and interpret both latent states and predictions.**

**Rating:** 7
**Confidence:** 3

**Review:**

### Strengths
- The paper is well-written with clear illustrations.
- The motivation is clear.
- The proposed method appears to be novel.

### Questions
- Why not apply the proposed method to existing public datasets? Can you provide the results?

---

### Official Review · Reviewer_oTtQ · 2023-06-15
**a concise algorithm for interpreting latent states**

**Rating:** 6
**Confidence:** 3

**Review:**

This paper presents a method to interpret latent states using highly related input features, interpret prediction using subsets of input features via latent rates and interpret changes in latent states over time.

The topic is interesting and the proposed method is straightforward and easy to follow. However, very few empirical results are presented in the paper. It would be better to include more evaluation, examples and discussion to show the efficacy of the proposed method.

---

### Meta-Review · Area_Chair_Nf7y · 2023-06-19

**Recommendation:** Accept (Poster)
**Confidence:** 4

**Metareview:**

Reviewers are generally positive in recommending the acceptance of this manuscript but also raise  concerns. Please address them in the final version.

---

### Decision · Program_Chairs · 2023-06-20

Accept (Poster)